# Assessment of Food and Waterborne Viral Outbreaks by Using Field Epidemiologic, Modern Laboratory and Statistical Methods—Lessons Learnt from Seven Major Norovirus Outbreaks in Finland

**DOI:** 10.3390/pathogens10121624

**Published:** 2021-12-14

**Authors:** Aleksandra Polkowska, Sirpa Räsänen, Pekka Nuorti, Leena Maunula, Katri Jalava

**Affiliations:** 1Health Sciences Unit, Faculty of Social Sciences, Tampere University, 33100 Tampere, Finland; apolkowska84@gmail.com (A.P.); pekka.nuorti@tuni.fi (P.N.); 2Pirkanmaa Hospital District, 33520 Tampere, Finland; sirpa.rasanen@tampere.fi; 3Department of Food Hygiene and Environmental Health, Faculty of Veterinary Medicine, University of Helsinki, 00100 Helsinki, Finland; leena.maunula@helsinki.fi; 4Department of Mathematics and Statistics, Faculty of Social Sciences, University of Helsinki, 00100 Helsinki, Finland

**Keywords:** norovirus, statistics, mathematics, disease outbreaks, cohort studies, fresh produce, Finland

## Abstract

Seven major food- and waterborne norovirus outbreaks in Western Finland during 2014–2018 were re-analysed. The aim was to assess the effectiveness of outbreak investigation tools and evaluate the Kaplan criteria. We summarised epidemiological and microbiological findings from seven outbreaks. To evaluate the Kaplan criteria, a one-stage meta-analysis of data from seven cohort studies was performed. The case was defined as a person attending an implicated function with diarrhoea, vomiting or two other symptoms. Altogether, 22% (386/1794) of persons met the case definition. Overall adjusted, 73% of norovirus patients were vomiting, the mean incubation period was 44 h (4 h to 4 days) and the median duration of illness was 46 h. As vomiting was a more common symptom in children (96%, 143/149) and diarrhoea among the elderly (92%, 24/26), symptom and age presentation should drive hypothesis formulation. The Kaplan criteria were useful in initial outbreak assessments prior to faecal results. Rapid food control inspections enabled evidence-based, public-health-driven risk assessments. This led to probability-based vehicle identification and aided in resolving the outbreak event mechanism rather than implementing potentially ineffective, large-scale public health actions such as the withdrawal of extensive food lots. Asymptomatic food handlers should be ideally withdrawn from high-risk work for five days instead of the current two days. Food and environmental samples often remain negative with norovirus, highlighting the importance of research collaborations. Electronic questionnaire and open-source novel statistical programmes provided time and resource savings. The public health approach proved useful within the environmental health area with shoe leather field epidemiology, combined with statistical analysis and mathematical reasoning.

## 1. Introduction

Food- and waterborne outbreaks caused by viruses are relatively common, especially during cold months [1]. Norovirus represents the highest burden of gastrointestinal infectious illness [2,3]. Recently, due to its high prevalence, norovirus has been estimated to cause the highest number of foodborne deaths in the UK [4]. Approximately 50–100 larger food or waterborne outbreaks are reported in Finland annually. Half of the notified foodborne outbreaks are caused by norovirus in Finland; however, during the COVID-19 pandemic, the number of norovirus outbreaks declined [5]. Norovirus spreads by several modes of transmission. Food- and waterborne viral outbreaks occur frequently [6,7,8], often followed by person-to-person transmission [9], including explosive outbreaks especially in healthcare institutions [10,11]. Symptomatic food handlers [12] or contaminated fresh produce (or similar) [13,14] are common foodborne means of transmission and vehicles. The common foodborne vehicles, oysters and frozen berries, are still causing a notable number of norovirus outbreaks globally [3,7,15]. Fresh produce is causing ever growing number of outbreak, including norovirus outbreaks [16,17]. Novel vehicles and means of transmission, like lake water and airbome transmission via faulty air ventilation valve, have been described recently [18,19]. Norovirus cases sharply declined by the social distancing measures and hand hygiene recommendations implemented with COVID-19 restrictions [20]. A possible increase in norovirus case loads after relaxation of COVID-19 pandemic social distance measures may remain to be observed [20,21].

Norovirus infections are diagnosed by reverse-transcription (RT)-PCR testing from faecal samples. Genogroup I is a common norovirus type in food- and waterborne outbreaks, while genogroup II, and especially genotype GII.4, is more common in person-to-person transmission [22,23]; however, this is not categorical. However, diagnostics take at least 24–48 h, and the Kaplan criteria are often used to assess the causative agent [24]. The Kaplan criteria require an incubation period of 24–48 h with an illness duration of 12–60 h [25]. In many countries, suspected food- and waterborne outbreaks are reported to national authorities, including in Finland [5], and further pathogen strain characterisation may be conducted [26,27]. Local health and environmental departments often need to resolve these outbreaks with limited resources, often under substantial media interest.

However, in many places, laboratory capacities are limited or laboratory confirmation of norovirus is not possible. Thus, to determine if the outbreak is likely caused by norovirus, clinical and epidemiological criteria can be used. Traditionally, the Kaplan criteria are used and include: proportion of vomiting >50%, mean incubation period 24–48 h, mean duration of illness 12–60 h and no bacterial pathogens detected [26]. We summarised the clinical presentation, epidemiological and microbiological features and recommendations of seven major norovirus food- and waterborne outbreak investigations conducted in Southwestern Finland between 2014 and 2018. The Kaplan criteria for norovirus outbreaks were evaluated and recommendations were proposed to effectively investigate localised, smaller-scale outbreaks.

## 2. Results

### 2.1. Descriptive Epidemiology

Altogether, 21.5% (386/1794) persons fulfilled the case definition in the seven cohort studies among 1794 persons enrolled; see Table 1. There were 155 secondary cases, 32 respondents travelled abroad one week before the illness and 47 respondents had other exclusion reasons, totalling 2028 respondents. Norovirus was the causative agent in all seven outbreaks, based on approximately five faecal samples tested per outbreak. Thus, samples from all the cases were not tested for noroviruses, nor was there information on all cases tested systematically. Additionally, sapovirus (1 isolate) was isolated in Outbreak_5, and rotavirus (3 isolates) and campylobacter (1 isolate) in Outbreak_6 as minor pathogens. No genotype determination was done in all norovirus-positive samples. Nine kitchen staff members were norovirus-positive (Outbreak_1,2,4,5). By excluding Outbreak_6 because of an undefined cohort, the attack rate was 41.6% (142/341). Symptoms were typical of gastrointestinal viral infection: vomiting 80% (293/368), diarrhoea 67% (235/352) and bloody diarrhoea 3% (4/124). Moreover, general symptoms of fever 43% (136/314), nausea 91% (334/366) and stomach pain 83% (296/356) occurred.

The symptoms by age group are presented in Table 2. Vomiting (96%, 143/149) and stomach ache (90%, 120/134) were more common in children, and diarrhoea was twice as common among the elderly (92%, 24/26) versus children (55%, 69/125). The mean duration of illness was longer in the elderly (63 h) versus children (38 h). The epidemic curve shows the distribution of the onset of symptoms; see Figure 1. The median incubation period was 38 h (3.25–96 h), and the duration of symptoms had a median of 31 h and mean of 42 h (0–318). The outbreaks mostly fulfilled the Kaplan criteria, except that the mean duration of symptoms was longer in Outbreak_5,7 and the mean incubation period was longer for Outbreak_1. After adjusting the proportion of symptoms by the Finnish population, the overall proportion of vomiting was 73%, the incubation period was 44 h and the duration of illness was 46 h.

### 2.2. Analytical Cohort Studies

The most likely vehicles were identified by the cohort study for Outbreak_1–6; the summary of these outbreaks is presented from this study or previously published studies in Table 1. Overall, three of the outbreaks were caused by an infected food handler (Outbreak_1,2,4), three by fresh produce transmission (Outbreak_2,3,4), one was suggested to be airborne (Outbreak_5), one was lake-water-associated (Outbreak_6) and one was transmitted person-to-person (Outbreak_7). The flow of the outbreak investigation during a typical food- (or water-) borne outbreak is presented in Figure 2. Additional to classical epidemiological methods, logical judgment to justify the role of the vehicle by reasoning was used. Cheese rolls were immediately suspected as all cases were explained with this exposure in Outbreak_2. The exclusion was used to restrict the investigation to those with a possibility of becoming exposed. Outbreak_3 consisted of two functions where the implicated strawberries were consumed and cases detected. The correction was used to clarify menu items and food sources. The information about the use of ice cubes in drinks and water was corrected for Outbreak_5. Food items containing strawberries from two functions were combined in Outbreak_3. Strengthening of the hypothesis was observed by the existence of contaminated ice cube machines in the same space as leaking air ventilation and observed dose–response on the amount of ice consumed (Outbreak_5) and intensity of contact with lake water (Outbreak_6).

### 2.3. Kitchen Inspections and State of Health of the Kitchen Staff

Hygienic inspections were conducted in preparatory kitchens with Outbreak_1–5,7 within 24–48 h post-notification. In all of the facilities, no major deviations from the food hygiene standards were observed. Moreover, guidelines were not disobeyed. No symptomatic food handlers were identified. However, a norovirus-positive food handler was identified in Outbreak_1,2,4,5, and for Outbreak_1,2,4, the food handlers were considered the most likely source of infection during the outbreak. In Outbreak_1,2, infected food handlers were asymptomatic during the outbreak. A norovirus-positive food handler was considered the source in Outbreak_1,2. One was all asymptomatic (Outbreak_2) and the other had been asymptomatic for three days (Outbreak_1).

### 2.4. Food, Water and Environmental Samples

Food samples from three outbreaks (Outbreaks_1,2,3) were analysed for noroviruses. Two food samples, cucumber–cheese rolls and lettuce–strawberry–vinegar salad, regarding Outbreaks_2,3, were further analysed. The latter was demonstrated to contain the norovirus genogroup II genome with a modified ISO method. Only low signals (Cq 35.99, and 34.69) were obtained in one of the parallel samples, in two consequent tests. One was obtained from the vinegar sauce sample after PCR inhibitor removal treatment and one from a salad leaf–vinegar sauce sample without PCR inhibitor removal treatment. Despite repeated efforts and multiple methods used, the presence of the norovirus genome could not be shown in the cucumber–cheese rolls in Outbreak_2. None of the water samples from Outbreaks_3,5,6 were positive for norovirus. Environmental samples from Outbreaks_2,4 were negative for norovirus. The potential role of the air ventilation valve was suggested during one of the inspections for Outbreak_5 caused by ice cubes.

### 2.5. Control Measures to Prevent Further Cases

All norovirus-positive and symptomatic food handlers were instructed to be withheld from risk work for appropriate periods (2 days minimum) in Outbreaks_1,2,4,5. As an asymptomatic, norovirus-positive food handler (Outbreak_2) and one food handler deferred from work for the required period but previously symptomatic (Outbreak_1) were considered likely sources, we recommend considering more stringent measures with food handlers (5 days away from high-risk duties). All suspected foods were immediately banned from use for all suspected foodborne outbreaks. In Outbreaks_2,5, however, many subsequent functions serving food in these commercial establishments had already taken place. Guidelines to improve hygiene as given during hygienic kitchen inspections were effective control measures in Outbreaks_1–5. As targeted measures, having only their own food during mass exams was recommended for students due to Outbreak_2, washing of fresh strawberries was discouraged in Outbreak_3, and reconstruction of the air ventilation valve was performed following Outbreak_5. Decontamination of the environment with Outbreaks_2,4,5 was ensured, with enhanced stringency of hygienic measures and hand hygiene in general. As exposure to one of the lakes in the Tampere area by beach visits was immediately noticed, a common factor among persons in major Outbreak_6, closure of the implicated beaches was implemented essentially until the end of the swimming season in 2014. We formulated our main experiences as key public health actions to be taken during foodborne outbreak investigations in the environmental health context; see Table 3.

## 3. Discussion

We investigated in detail seven major viral gastroenteritis outbreaks in Southwestern Finland between 2014 and 2018 among a total of around 400 outbreaks, resolved the causative agents and traced food and environmental vehicles, identified infected food handlers and implemented necessary control measures. Clinical information was used as initial evidence about the causative agent, and the Kaplan criteria were assessed. Vomiting was more common in children than in older patients, whereas the elderly had often prolonged symptoms and diarrhoea up to 14 days. The contamination most likely occurred in the food preparation premises for all foodborne outbreaks, and infected food handlers were an important source in this limited number of viral foodborne outbreaks. We used rapid and stringent epidemiological, microbiological, statistical, mathematical and circumstantial evidence to identify environmental vehicles to drive the decision-making process. Guidelines are as follows. (1) Strong epidemiological evidence includes a statistical association in a well-conducted analytical epidemiological study, or convincing descriptive evidence. Examples of convincing descriptive epidemiological evidence are provided in Appendix A. (2) Strong microbiological evidence includes the identification of an indistinguishable causative agent in a human case and in a food, a food component or its environment, which is unlikely to have been contaminated coincidentally or after the event, or the identification of a causative agent, such as a toxin or bio-active amine, in the food vehicle, in combination with clinical symptoms and the onset of illness in outbreak cases strongly indicative of/pathognomonic to the causative agent. (3) Comprehensive product-tracing investigation can provide strong evidence in case a common point along the food production and distribution chain is identified for all or a large proportion of cases who were exposed and for whom a place of exposure/point of sale could be identified; see Appendix A. Help from the general public and gaining their trust was the ultimate key to success in these outbreak investigations.

The clinical presentation was typical of norovirus infection. The incubation period for vomiting varied from 4 h to 4 days. The maximum was in line with what has been observed before [28], but the minimum of 4 h is somewhat shorter than observed before [28], but this was not verified by the patient and remains thus uncertain. The median length of symptoms of 31 h was within the reported range (12–60 h). For the longer duration of illness in the elderly patient, there are many reasons, such as lowered immunity. Although reported earlier, further studies are needed for the observations of children having more vomiting and stomach aches, while the adults and elderly had more diarrhoea [28,29]. A longer duration of illness in the elderly is consistent with the longer shedding of norovirus in the elderly [30]. The children <5 years of age had a somewhat longer duration of illness in this study, as observed in [31,32], albeit shorter than in the elderly. This needs to be taken into account when assessing a presumptive norovirus outbreak—the age distribution of the cases should be reflected in the assessment.

The significance of the Kaplan criteria has diminished over the years as rapid PCR-based methods are used to detect norovirus in faecal samples. However, they still have an important position in the early stages of the investigations. Much of the crucial public health action needs to happen prior to 48 h, the typical time required to process faecal samples. For the lake-water-associated Outbreak_6, as norovirus was considered an unusual agent for a lake-water-associated outbreak, a lot of speculation occurred during the first 48 h about the causative agent, both within the outbreak team and the media. More refined criteria than Kaplan’s have also been suggested [27], but the level of detail in the beginning did not permit sophisticated analysis with any of the outbreaks in the current study. Moreover, the media informed the outbreak team about Outbreak_2, and an extremely rapid risk assessment took place based on initial symptoms, i.e., using Kaplan criteria, to begin with. Overall, the media interest usually ensures effective reporting during outbreaks [33] and this was also the general experience during this study.

Immediate control measures and guidance on hygienic practices were implemented to prevent further cases. Hygienic inspections among food operators efficiently corrected actions as in [34,35]. The withdrawal of two days from work since the clearance of gastrointestinal symptoms is a commonly used measure [36,37], and there are presumably constraints to lengthen this. However, we recommend enforcing voluntary withdrawal of these persons for at least 5 asymptomatic days and very strict hand hygiene after this period from risk duties as, in one of the outbreaks in the current study, one kitchen staff member likely caused the outbreak after three asymptomatic days post-illness. The irony is that performance of kitchen hygiene may be excellent, but, due to the persistence and virulence of norovirus, it may spread through kitchen staff despite the hygienic procedures in place. Our overall impression was that if a person is actively transmitting norovirus, it may be close to impossible to prevent the spread of the virus to the environment and surroundings.

Food and environmental samples often remain negative [38], especially if the source is an infected symptomatic or asymptomatic food handler [12,37]. The amount of viruses needed to be detected from food and environmental samples is much higher than needed to cause an infection. Contamination often happens during the cutting or handling of fresh produce [14], which was mostly also the experience during the current study for Outbreak_1–4. Based on epidemiological evidence of Outbreak_2, cucumber slices were considered biologically the most plausible. In recent years, improved devices for the removal of RT-PCR inhibitors from frozen soft fruit have been reported [38,39] and they were found useful in this study. Although further genotyping of the virus in food was unsuccessful (not described), the noroviruses detected in the food and in patients represented genogroup GII, which does not contradict the notion that the strawberry salad was the likely source of Outbreak_3. The low positive norovirus signals obtained in the small and long-stored salad sample could have been caused by cross-contamination between the other salad ingredients (leaves, vinegar sauce) and strawberries.

Classical, robust epidemiological data analysis during these outbreak investigations was used. Data are to be described carefully in terms of time, place and person [40], and this was performed repeatedly during these investigations [17,18,19] (this study). Defining the cohort was one of the challenges. With a city-wide outbreak (Outbreak_6), a voluntary enrolment process was used, as there was no list of public beach visitors. This may have introduced selection bias [41], but this was considered the best option available. Electronic questionnaires were mostly used. However, environmental health officers and nurses also interviewed the elderly and child patients by phone (detailed data not recorded). It is recommended to include children in the analytical studies as they remember well what they have eaten at school (the typical occurrence) [42]. The non-commercial Kobo system was valuable and easy-to-use; however, it is recognised that there are plenty of options for various needs. Electronic data collection devices have become widely used during outbreak investigations [43]. Moreover, the use of loyalty card information may be valuable and has been extensively used, especially with hard-to-find vehicles, such as listeria [44].

Mathematical reasoning and the “thinking-outside-of-the-box” approach were exploited for resolving two of the major outbreaks in the current study, presenting rare or novel vehicles of lake water (Outbreak_6) and a series of unlucky constructional elements leading to the airborne contamination of ice cubes through a faulty air ventilation valve (Outbreak_5) [18,19]. Despite the usefulness of epidemiological data, advanced statistical and other data methods are often needed in outbreak situations [40,45]. Retrospective cohort studies are less prone to bias (than case–control studies), but high attack rates are theoretically required [46]. Fudging the numbers was performed. The post-COVID-19 era may see an increase in norovirus outbreaks once social distancing is relaxed, as norovirus cases generally declined during the pandemic [47]. A rigorous public health approach was fruitful, as has been also detected by others [48], and should be fully applied to environmental public health as well in order to ensure adequate public health action. This included, in particular, determining the mechanism behind the outbreak’s occurrence, particularly how the food became contaminated (Outbreak_5, identification of faulty air ventilation valve contaminating ice cubes) or how patients became infected (Outbreak_6, role of lake water for the infections).

The decision-making process is challenging in outbreak investigations [39,45]. This includes, more specifically, the challenges for the decision-maker to be able to bridge gaps in often patchy information and ongoing investigations, while the need to optimise interventions remains an ever-growing challenge [40]. Outbreak investigations should also move away from a purely acute response towards a more phased response with preparation, response and recovery. Outbreak investigations are also often challenged by the general public’s reluctance to comply, and more social science and diplomatic skills are needed [45]. In foodborne outbreak investigations, decision-makers need to prohibit the use of suspected food items as a precautionary principle. Large batches of foods may need to be banned due to patchy information in the beginning. This may create friction between authorities and food operators; however, given the experience of the current study, the main concern for food operators is whether their name is released to the public. Often, the value of implicated foods is minimal compared to the potential image damage due to negative media attention. Food control authorities work with food industry representatives during routine food premises inspections, but also during outbreak investigations when premises are inspected for possible faults that led to the outbreak. It is recommended that public and environmental health professionals should work constructively with the food industry as, often, the primary source is a real challenge to resolve (this study, unpublished listeria outbreaks). Obviously, this does not and must not jeopardise the integrity of the authority action. Legal enforcement may be needed, but should be kept as a separate issue, not to be directed against individual industry staff, and left possibly to the post-outbreak period as the collaboration may become more formal and non-negotiable. Legal enforcement is not part of the public health outbreak response, and serves best as a warning sign for other key stakeholders in the food industry. Authoritative power and some legal enforcement were used in most of the outbreaks described in this study. It was used to prohibit the use of ice cube machines until machines were cleaned and the air ventilation valve relocated (Outbreak_5), to prohibit swimming in the four implicated lakes (Outbreak_6), to recommend that only students’ own food should be eaten at important mass exams (Outbreak_2) and to discourage the practice of washing strawberries (Outbreak_3). Hand hygiene was continuously promoted. Some limitations of the study were the lack of knowledge on the source and size of the population affected. The studies were also dependent on a few key experts, with little replacement in times of absence. Major limitations of all outbreak investigations were a lack of staff and resources to pay for the microbiological analysis of food samples as departmental budgets were tight. Close collaboration with research institutes was crucial.

## 4. Materials and Methods

### 4.1. Description of the Events and Patients

Out of 400 suspected food- and waterborne outbreaks that occurred in Southwestern Finland between 2014 and 2018, seven major viral gastroenteritis outbreaks were selected, where a comprehensive epidemiological study was undertaken due to a high number of cases, media attention and unknown vehicles, potentially of wider distribution. Details on time, place, person, Kaplan criteria, vehicle, source of evidence, genotype and kitchen staff status of each outbreak are summarised in Table 1. The summary descriptions of the four unpublished outbreaks are given in Appendix A and for the published Outbreak_4–6 in [17,18,19]. Faecal samples were collected from ~5 patients per outbreak and tested for routine pathogens, including norovirus.

### 4.2. Cohort Studies

To evaluate the Kaplan criteria, individual participant data meta-analysis with data from all seven outbreaks was conducted; see Appendix A. The cohort population included those attending the implicated event. The risk foods for each implicated event were analysed for each outbreak separately as vehicles were unique and different from each other. For descriptive analysis, the clinical response from all those enrolled was combined. A case was defined as Outbreak_1–7 with participants with diarrhoea, vomiting or two other symptoms after the implicated event. Secondary cases within the households were excluded; only one primary case or cases with the same onset date per household were included. Outbreaks complying with four Kaplan criteria to distinguish norovirus outbreaks by clinical presentation were compared. Statistics Finland population estimates (2018) were used to calculate adjusted total Kaplan criteria by age-dependent population fractures. Crude proportions were taken proportionally to the age-dependent population fraction.

Electronic questionnaires with KoboToolbox for Outbreak_1,2,3,5,7 [49] were created; see Appendix A. An example questionnaire is provided online https://ee.kobotoolbox.org/x/#iicYWqgG (accessed on 13 December 2021). A paper-based questionnaire was used for Outbreak_4 and a local electronic form for Outbreak_6. Participants were given a few days to a week to respond to the self-administered online questionnaire, with reminders if needed. Standard demographic questions and clinical symptoms were asked, including foods and drinks consumed at the functions, and swimming and beach activities for the Outbreak_6.

### 4.3. Descriptive and Statistical Analysis

The results were analysed using LibreOffice Calc (Linux), Excel^®^ and R (Cran). The proportions for gender, age and symptoms of the cases for the enrolled were calculated. While performing the epidemiological investigations, it was also ensured that cohort members had the possibility of becoming exposed by including only those participating in the event serving the implicated food. We combined variables for commonly served foods, especially for water, and used fudging by adding one person to all four cells if all cases were exposed. We calculated the incubation period between exposure of interest and onset of illness for any symptoms (excluding Outbreak_6) and the duration of illness (from the onset of symptoms to the end of symptoms) to assess Kaplan criteria. Symptoms were adjusted by the population fractures. For the risk ratios, profile likelihood-based confidence intervals were used. The R code for installation and calculating the risk ratios and respective confidence intervals is available in Appendix A; moreover, the Epidemiologist R Handbook may be useful (https://epirhandbook.com/en/ (accessed on 13 December 2021)). Furthermore, circumstantial evidence is often used in epidemiological studies to support the hypothesis [50,51]. This was formulated further and we used four mathematical approaches: logical judgment, exclusion method, correction and strengthening of evidence [52,53,54].

### 4.4. Kitchen Inspections and Environmental Sampling

The menu and food items, food preparation recipes and associated documents were obtained from kitchen staff members. Food preparation kitchens and premises were inspected for Outbreak_1–5,7. Food preparation facilities, room spaces, practices, instructions and guidelines were evaluated using a routine protocol for conducting hygienic inspections [55], with a focus on the foods prepared for the function. Meal menus and preparation practices were evaluated. The kitchen staff were interviewed, including abdominal symptoms prior to and after the outbreak. Available fresh produce and water were sampled for testing using routine methods, as previously described [17,18,19].

### 4.5. Environmental Microbiology

The environmental samples of Outbreaks_4,5,7 were analysed as previously described [17,18,40]. Food samples linked to Outbreaks_1,2,3 were analysed for noroviruses according to ISO standard (ISO 15216). In addition, food samples linked to Outbreaks_2,3 were subjected to further virus analyses at the Department of Food Hygiene and Environmental Health, University of Helsinki. These foods consisted of cheese–cucumber rolls (Outbreak_2) and salad (iceberg lettuce and strawberry residues in vinegar sauce; Outbreak_3).

In Outbreak_2, virus extraction was performed by rinsing the cucumber slices available (much less than 25 g recommended) that were inside the rolls, after which extracts of the samples, namely 1.5 mL and 2.0 mL of the liquid, were directly subjected to nucleic acid extraction. Nucleic acids were treated with the OneStep PCR inhibitor removal kit (Zymo Research, Irvine, CA, US). After storage for two years at −20 °C, the cheese rolls were re-analysed using two other virus extraction methods described by [56]. Cheese and cucumber slices were combined as one sample and rinsed with sparkling water according to Method 1 [56]. Ice was scraped from the frozen rolls as well as collected from the roll bags. Direct RNA extraction with PEG supplement as described in Method 2 was performed for the ice samples.

In Outbreak_3, the following methods were applied for the virus extraction: (1) 1 mL salad vinegar sauce was directly subjected to RNA extraction, or (2) the ISO 15216 method was used with minor modifications. Approximately 25 g salad (vinegar sauce only or salad leaves and vinegar sauce) was taken and viruses were eluted from them with 40 mL TGBE buffer. Then, PEG precipitation and butanol–chloroform phase extraction were used according to ISO 15216. The extracted nucleic acids with or without treatment with OneStep PCR inhibitor removal kit were used as templates in RT-PCR. After storage for two years at −20 °C, the salad was analysed again using the two rapid methods described by [56]. One sample consisted of salad leaves that were rinsed with sparkling water and treated as described in Method 1 [56]. Two other samples consisted of vinegar sauce with small pieces of salad in it, and ice flakes from the salad container. These two samples were analysed using the direct RNA extraction method with PEG supplement, as described in Method 2. Nucleic acid extraction of all samples was performed using NucliSens reagents with MiniMAG apparatus (Biomerieux, Marcy l’Etoile, France). RT-PCR was performed with QuantiTect probe RT-PCR kit (Qiagen, Hilden, Germany) and Rotorgene PCR cycler. Primers and a FAM-labelled probe were used according to ISO 15216 [57].

## 5. Conclusions

We established a novel public health approach for investigating public health events, implemented in an environmental health context when attempting to resolve food- or waterborne outbreaks in local settings with limited resources. We acknowledged the challenges of local authorities when faced with unprecedented events and attempts in resolving complex outbreaks. We aimed to develop tools and methods to aid in these situations. We recommend using holistic epidemiological, microbiological, mathematical and statistical approaches to resolve point-source food-, waterborne and environmental outbreaks. The clinical Kaplan criteria were useful in the early stages of the suspected norovirus outbreaks by age profiling of the early cases, as vomiting and fever were more common among children, while the duration of illness, especially diarrhoea, was longer among the elderly. Immediate, joint and multiple field visits to the food preparation premises by the food control officers and epidemiologists addressing public health action are vital and recommended. The freely available electronic data collection and statistical programmes were useful, and many easy-to-use manuals are available online. Initiatives developing microbiological methods for new environmental matrices with enhanced sensitivity for viral detection are to be supported. Kitchen staff should be rigorously withheld from work with any gastrointestinal symptoms for 48 h, and preferably kept from high-risk food duties for a longer period. Food premises inspections within 24–48 h should take place, as well as implementing immediate control measures, launching and analysing the cohort study rapidly and simultaneously releasing internal and external press releases. We recommend asking for help from the general public on their observations, participating in the studies and informing their close contacts about potential threats.

## Figures and Tables

**Figure 1 pathogens-10-01624-f001:**
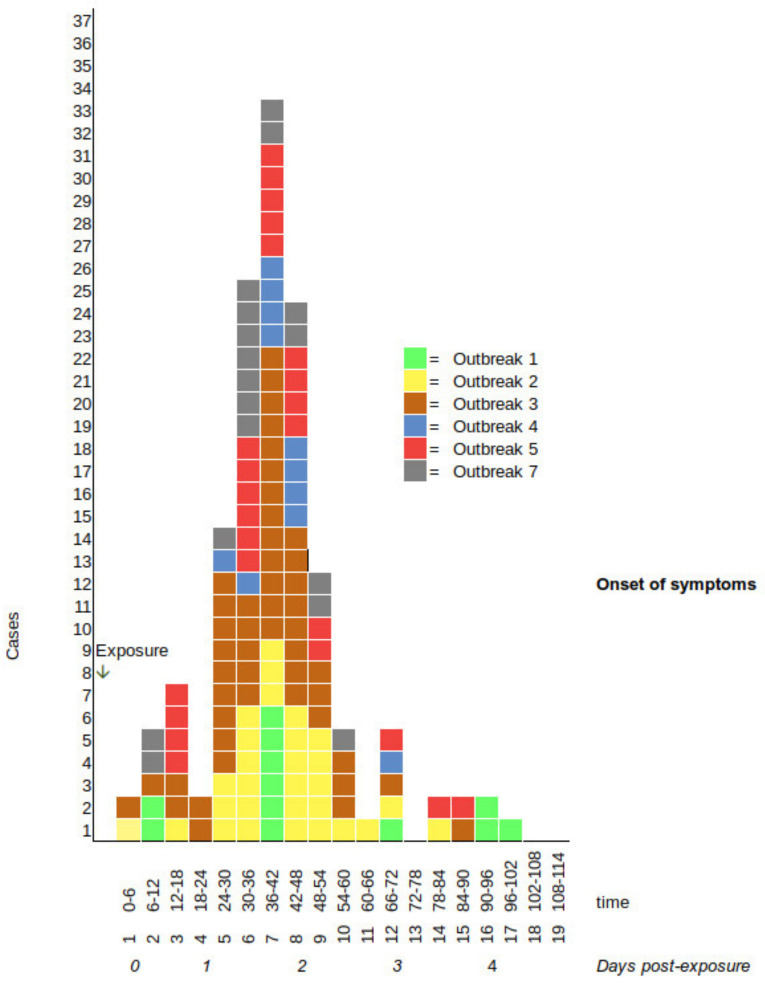
Epidemic curve by the onset of illness and exposure timing for Outbreak_1–7.

**Figure 2 pathogens-10-01624-f002:**
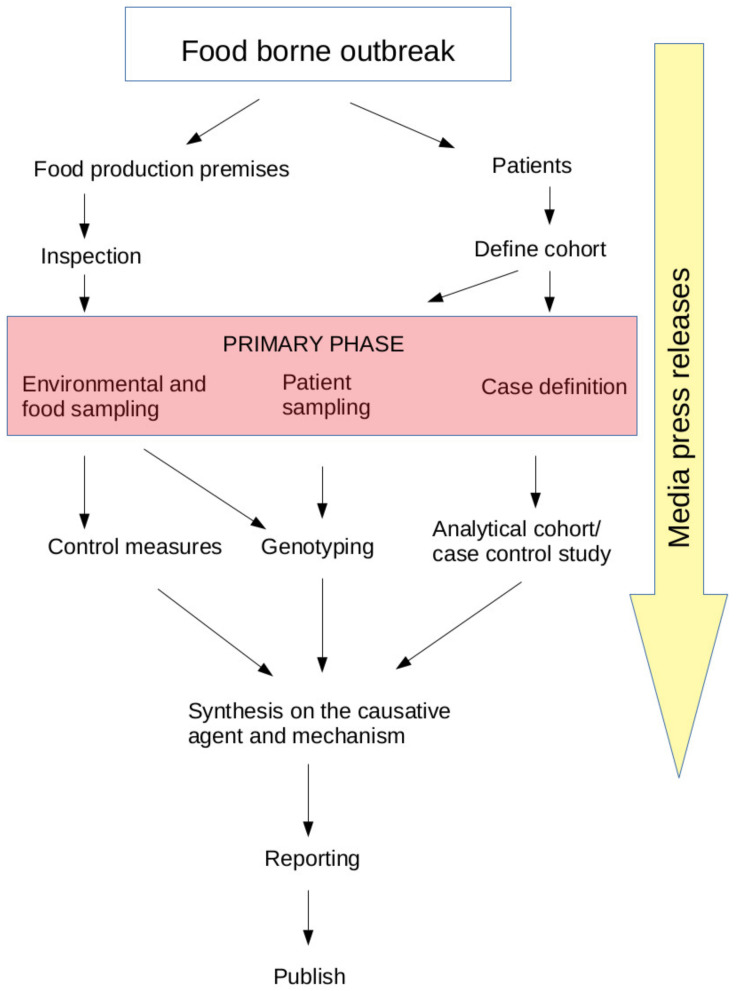
Schematic representation of the outbreak investigation.

**Table 1 pathogens-10-01624-t001:** A detailed list of selected major viral gastrointestinal, non-institutional outbreaks in Southwestern Finland during 2014–2018.

Outbreak Name	Time	Place	Number of Enrolled (Included in the Cohort Study)	Number of Cases Enrolled in the Cohort Study	Vomiting, % (No./Total)	Mean Length of Incubation Period, hours	Mean Length of Duration of Symptoms, hours	Vehicle (RR, 95% Confidence Intervals)	Evidence	Norovirus Genotype(s)	Infected Food Handler (If not Considered the Source)
Outbreak_1 (this study)	21–23.3.2018	Birthday party (catering service)	51 (38)	12	67% (8/12)	51	43	Mini-burgers, RR = 5.0 (95% CI 1.58–29.81)	Cohort study, Patient samples	GII.P7	Yes
Outbreak_2 (this study)	7.2.2017	School, institutional kitchen	83 (82)	29	66% (19/29)	41	35	Sliced cucumber (within cheese rolls), RR = 6.12 (95% CI 1.54–103.89)	Cohort study, Patient samples	GII	Yes
Outbreak_3 (this study)	4–5.3.2017	Catering service, two functions	72 (72)	49	73% (35/48)	37	49	Washed, fresh strawberries (cream cake and salad), RR = 2.28 (95% CI 1.14–6.64)	Cohort study, Patient samples, Food samples	GII	No
Outbreak_4 [17]	26.1.2015	Lunch restaurant	27 (21)	11	82% (9/11)	38	28	Salad buffet, RR = 2.33 (95% CI 1.00–8.41)	Cohort study, Patient samples	GI.P7	(Yes)
Outbreak_5 [18]	9–10.12.2016	Evening restaurant, two functions	102 (91)	24	58% (14/24)	39	61	Ice cubes, air contamination through faulty air ventilation valve, RR = 6.46 (95% CI 1.46–113.97) to RR = 8.17 (95% CI 1.67–145.52)	Cohort study, Patient samples	GI and sapovirus	(Yes)
Outbreak_6 [19]	21–28.7.2014	Four fresh water lakes	1656 (1453)	244	85% (193/227)	–	38	Contaminated lake water, head under water, RR = 3.24 (95% CI 2.31–4.71)	Cohort study, Patient samples	Several noroviruses: GI, GII,	No
Outbreak_7 (this study)	22.10.2016	Private christening party (catering service)	37 (37) (2 children <2 years were excluded from analysis)	17	88% (15/17)	37	66	Person-to-person transmission, not significant for any exposures	Cohort study, Patient samples	GII.P16-GII.4	No

**Table 2 pathogens-10-01624-t002:** Clinical symptoms in various age groups.

Clinical Characteristics	Children <18 Years of Age *n* = 150,% (Exposure/Total)	Adults 18-64 Years of Age *n* = 207, % (Exposure/Total)	Elderly 265 Years of Age *n* = 27, % (Exposure/Total)	Comparison of Children to Elderly, 2-Sided Test, *p*- Value
Diarrhoea	55% (69/125)	71% (141/199)	92% (24/26)	0.004
Vomiting	96% (143/149)	69% (132/192)	64% (16/25)	<0.001
Bloody diarrhoea	0% (0/7)	3% (3/96)	5% (1/21)	0.56
Nausea	96% (133/138)	88% (176/200)	88% (23/26)	0.086
Stomach ache)	90% (120/134)	80% (156/194)	69% (18/26)	0.0059
Fever	44% (50/113)	46 % (80/175)	24% (6/25)	0.062
Time between exposure and illness onset (mean)	64 h (*n* = 7)	39 h (n = 113)	36 h (*n* = 21)	0.19
Duration of illness (mean)	38 h (*n* = 142), in children <5 years 48 h, (*n* = 27)	43 h (*n* = 186)	63 h (*n* = 18)	0.013

**Table 3 pathogens-10-01624-t003:** Main key public health entities during a foodborne outbreak investigation.

Entity	Justification and Reasoning
Aim and scope	The immediate aim of the foodborne outbreak investigation is to prevent further illness in the community by withdrawing any suspect food items from the food chain during the initial steps of the outbreak investigation. As a long-term goal, the aim is to determine the complete mechanism by which food became contaminated during primary production and food processing and which factors contributed to the spread of the pathogens to cause human illness. Formulation of food safety standards and recommendations is a priority.
Speed	Most of the environmental public health action occurs within the first 24–48 h with successful outbreak investigations. If this important window of opportunity is missed, it may result in increasing number of new cases and potentially excess deaths. Media response also reflects the success and speed. The lead epidemiologist needs to visit quickly all outbreak-affected areas in person, communicate with the press and social media and conduct active case finding. As a rule of thumb, once the environmental health unit is notified of a potential outbreak, the situation is already severe and immediate action needs to be taken (excluding family outbreaks, etc.).
Formulation of multidisciplinary outbreak control team (OCT)	OCT needs representatives from public and environmental health, clinical medicine, statistics and media communications. A formal lead to present in the media and an epidemiologist responsible for practical lead of the investigations are needed and preferably should be two separate persons.
Rapid food facility assessment	Environmental health will lead a thorough assessment of the implicated facility to formulate hypothesis on the causative agent and mechanism of the outbreak. Often, several visits over the course of the first days of the investigation are needed.
Prevention of further spread	Immediate risk assessment; communication with all colleagues with knowledge on the topic. Prevention of possible contaminated food and identification of possible infected food handlers. Downplay of any isolated leading of the incident. Use of Kaplan criteria to assess the possibility of norovirus outbreak. Lead epidemiologist is at the service of others, not vice versa.
Collaboration with research institutes	Food and environmental samples present as challenging matrix for microbiological and genetic analysis. Outbreaks present perfect opportunity to develop analytical methods further with research institutes. This also applies to theoretical aspects of epidemiological and statistical methods.
Epidemiological and statistical methods	Outbreak response and determination of the implicated vehicle and causative pathogen should be directed by likely probabilities of options based on rapid risk assessments, due to be updated during the course of the investigation. The environmental public health action is prioritised based on mathematical probabilities of events, based on literature and past experience.
Technical advances	Novel technologies aid in making outbreak investigations quicker, more reliable and robust. These include, e.g., electronic, online questionnaires and open-source statistical and mathematical programs (R, Python, etc.). In addition, use of consumer purchase data (loyalty cards, etc.) may be useful.

## Data Availability

No other data than those provided in this manuscript and associated files are available. None of the data are publicly available and were part of local outbreak investigation in Southwestern Finland.

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
