# Peer review of "Assessment of Food and Waterborne Viral Outbreaks by Using Field Epidemiologic, Modern Laboratory and Statistical Methods—Lessons Learnt from Seven Major Norovirus Outbreaks in Finland"

_pathogens, 2021, doi:10.3390/pathogens10121624_

Round 1

Reviewer 1 Report

This is a well written report on assessment by a varity approaches and methods for the food and waterborne norovirus outbreaks.

The manuscript has a very solid content. The analysis method suggested by the authors will be of help in understanding the epidemiologic analysis of the outbreak of food and waterborne viral outbreaks and preparing an initial response plan. If there are any limitations of the study, please described it.

Author Response

Dear Reviewer,

Thank you for your positive feedback. As requested, the limitations section was expanded at the end of the discussion. 

Reviewer 2 Report

Minor

Line 47: ‘vehicle s’ remove space

Line 50: concerning the covid pandemic, I would like to see a reference showing that the social distancing had also an impact on norovirus.

Line 57: ‘duration of …’ sentence is not finished

Fig 1: labeling of x-as should come next to the values not below

Overall: check spaces and double spaces remove were necessary

Overall: check points at the end of sentences

Author Response

Dear reviewer,

Thank you for your positive feedback

Comments and response:

  • line 47 (and other places) extra spaces were removed (cracks were unfortunately introduced by the referencing programme, apologies).
  • l. 50 Covid pandemic and social distancing: A reference has been added.
  • Figure 1 was updated 
  • spaces checked, double spaces removed and punctuation checked.

Reviewer 3 Report

The manuscript from Polkowska, A. et al. describes in a very detailed manner the holistic combination of epidemiological, mathematical and related approaches used to face recent human Norovirus outbrakes in Finland. The manuscript is well written, correctly structured, the methodology is correct, the bibliography seems to be adequate and the content is relevant for the field. 

  1. There seems to be a mix-up with the tables in document. Table 3 (pg. 5) appears before Table 1 (pg. 10) in the text, and I could not find Table 2, although it is cited in the text (line 88). Table 3 would also benefit from some editing.
  2. Sentence in line 57 is unfinished….
  3. Some minor typos, like for example T he (46) and vehicle s (47)...

Author Response

Dear reviewer,

Thank you for your positive feedback

  1. Tables were restructured, table 2 added. The formatting of the document did not allow table 2 to be added as a text table (I hope editorial office may do the editing from figure to table as appropriate). Table 3 was restructured and some wording changed.
  2. Sentence was finalised.
  3. Cracks of words were corrected.